# Sudden Occurrence of Pacemaker Capture Failure during Irreversible Electroporation Ablation for Prostate Cancer in Post-COVID-19 Patient: A Case Report

**DOI:** 10.3390/medicina58101407

**Published:** 2022-10-07

**Authors:** Min Suk Chae, Nuri Lee, Hyun Jung Koh

**Affiliations:** Department of Anesthesiology and Pain Medicine, Seoul St. Mary’s Hospital, College of Medicine, The Catholic University of Korea, Seoul 06591, Korea

**Keywords:** COVID-19, electroporation, EMI, pacemaker, prostate cancer

## Abstract

Irreversible electroporation (IRE) ablation is a novel treatment option for localized prostate cancer. Here, we present a case of an abrupt and fatal arrhythmia during the IRE procedure in a prostate cancer patient with an implanted permanent pacemaker. A 78-year-old male patient with a pacemaker due to sick sinus syndrome and syncope was scheduled for IRE prostate ablation surgery under general anesthesia. He had a history of recovering from coronavirus disease 2019 (COVID-19) after having been vaccinated against it and recovered without sequalae. Pacemaker interrogation and reprogramming to asynchronous AOO mode was carried out before surgery, however, sinus pause occurred repeatedly during ablation pulse delivery. After the first sinus pause of 2.25 s there was a decrease in continuous arterial blood pressure (ABP). During the delivery of the second and third pulses, identical sinus pauses were observed due to failure to capture. However, the atrial-paced rhythm recovered instantly, and vital signs became acceptable. Although sinus pause recovered gradually, the duration thereof was increased by the delivery of more IRE pulses, with a subsequent abrupt decrease seen in blood pressure. The pacemaker was urgently reprogrammed to DOO mode, after which there were no further pacing failures and no hemodynamic adverse events. For patients with pacemakers, close cardiac monitoring in addition to the interrogation of the pacemaker during the electromagnetic interference (EMI) procedure is recommended, especially in the case of having a disease that may aggravate cardiac vulnerability, such as COVID-19.

## 1. Introduction

Irreversible electroporation (IRE) is a novel technology involving the induction of cancer cell death (the formation of micropores within cellular membranes, the alteration of the membrane shape, and activation of the apoptosis process) via very short but strong pulsed electric fields under medical imaging guidance, such as magnetic resonance imaging (MRI). Although surgical methods have remained a definite treatment for prostate cancer with a large mass or at a severe stage, nonsurgical efforts, such as an emerging focal therapy (i.e., IRE), have been gradually accepted as a tolerable and minimal oncologic control and have clinically relieved the perioperative stress burden that is involved in the removal of the whole prostate gland with radical prostatectomy or radiotherapy. The IRE has been considered a useful treatment option for localized prostate cancer that can minimize surgery-related side effects, such as erectile dysfunction and urinary incontinence [1]. During anesthetic care for IRE, it requires full muscular relaxation and constant cardiac surveillance to prevent the unexpected occurrence of rhabdomyolysis and arrhythmia [2].

Fatal arrhythmia due to strong currents may occur when IRE is performed in organs close to the heart, such as the liver, pancreas, and kidneys, so it should be performed with electrocardiography (ECG) monitoring and synchronization [3,4]. Because of the requirement of the electrical conduction (at least 1000 V) for efficient tumor ablation, Deodhar et al. suggested that the unsynchronized performance of IRE close to the heart (within 1.7 cm of the heart) may lead to adverse cardiac events. However, IRE gating of the cardiac cycle (i.e., synchronization) may attenuate the risk of myocardium susceptibility to arrhythmogenesis and fibrillation. Beyond the effect of remote distance, the effect of ablation on the myocardium may predominantly decrease to a minor arrhythmia level [4]. Contrary to previous liver, pancreas, and renal IRE reports, as the prostate is remote from the heart, there is little concern about arrhythmia, so most cases are performed without such considerations [5].

In patients with a pacemaker consisting of an impulse generator and leads to carry the electrical impulse to the heart, electromagnetic interference (EMI) with pacemaker function may occur intraoperatively and can lead to pacemaker failure and hemodynamic compromise [6,7]. Electric pulsed medical devices, such as a monopolar electrosurgery, may be one of causes of EMI that impair implantable cardioverter defibrillators and the risk of clinically meaningful EMI was higher in above-the-umbilicus noncardiac surgery (7%) rather than below-the-umbilicus surgery (almost 0%). This EMI finding persisted predominantly in above-the-umbilicus surgery despite protocolized dispersive electrode positioning [8]. For below-the-umbilicus surgery, such as IRE for prostate cancer, the meaningful EMI risk may be negligible which would indicate that perioperative alteration of cardiac management (i.e., suspending antitachycardia therapy) would be potentially unnecessary.

Nowadays, there have been reports that COVID-19 infection is related to cardiac arrhythmias such as prolonged QT interval [9,10]. Cardiac manifestation etiologies may be multifactorial and potentially involved in direct viral myocardial injury, hypoxia, hypotension, aggravating inflammation, ACE2-receptors downregulation, toxicity of COVID-19 treatment drugs, and endogenous catecholamine adrenergic status [11]. Acute myocarditis and ventricular arrhythmia may be the first clinical signs of COVID-19, but in the Italian epidemic, many nonhospitalized patients without or with mild symptoms were found dead at home while in quarantine. After hospital discharge without any respiratory or cardiac manifestations, the patients may have had latent myocardial scars that resulted in atrial or ventricular fibrosis and, subsequently, increasing the propensity of cardiac arrhythmia [11].

To the best of our knowledge, reports of patients with existing pacemakers being affected by arrhythmia after recovering from COVID-19 during IRE for prostate cancer, where it is remote from the heart, are rare [3]. Here, we present a case of abrupt and potentially fatal arrhythmia during IRE for prostate cancer in a patient with a permanent pacemaker due to sick sinus syndrome and syncope, who had a history of COVID-19 infection before the operation.

## 2. Case Presentation

A 78-year-old male patient (height of 175 cm and weight of 80 kg) was diagnosed with prostate cancer (adenocarcinoma, acinar, Gleason score seven, Grade group three) on tissue biopsy during a follow-up study of prostate hypertrophy. He was scheduled for NanoKnife (AngioDynamics Inc., Latham, NY, USA) electroporation and IRE ablation under general anesthesia instead of prostatectomy considering the age-related postoperative complication. In preoperative assessments, he was found to have sick sinus syndrome with frequent syncope and had undergone implantation of a pacemaker (Advisa DR MRI; Medtronic, Minneapolis, MN, USA) operating in AAI mode 7 years earlier (Figure 1).

Additionally, he had hypertension and dyslipidemia, for which he was regularly taking lercanidipine (the dihydropyridine class of calcium channel blocker), valsartan (angiotensin II receptor blocker), rosuvastatin (inhibiting HMG-CoA reductase), and dutasteride (a 5α-reductase inhibitor). There had been no typical symptoms during the 7 years of pacemaker use. He had a COVID-19 related history and had been vaccinated (Pfizer-BioNTech, New York, NY, USA) against COVID-19 six months before his scheduled IRE. Two months before the scheduled IRE, clinical COVID-19 was suspected with a sore throat, a mild fever (peak 37.3 °C with acetaminophen control),and a cough and finally a diagnosis with a positive test result of real-time reverse transcription–polymerase chain reaction (RT–PCR). During his quarantine at home, his infection course was tolerable without severe complications (such as dyspnea, hypoxia, and hypotension). After a week quarantine, he returned to daily life with a mild muscle ache and fatigue and did not show major cardiac or respiratory sequelae. The day before the IRE (post-COVID-19 infection day 60), the multidisciplinary team determined his physical status was tolerable for elective IRE procedure based on the mild initial infection, no ongoing symptoms of COVID-19, comorbid and functional status, clinical priority and risk of prostate cancer progression, and the complexity of IRE. The team did not recommend repetitive RT-PCR measures, because of the possibility of prolonged detection of COVID-19 particles [12]. In the preoperative assessments, laboratory variables were within the normal limits (sodium, 138 mmol/L; potassium, 4.1 mmol/L; and calcium, 9.1 mg/dL). Troponin-T 0.0105 ng/mL, N-terminal pro-brain natriuretic peptide (NT-proBNP; 71.8 pg/mL), and creatinine kinase muscle brain (CK-MB; 1.71 ng/mL) were also within the normal limits. An ECG revealed a heart rate of 63 beats per minute (bpm), a regular sinus rhythm, and no pacemaker dependency. In a preoperative two-dimensional echocardiography, the left ventricular systolic function did not show specific abnormalities; the ejection fraction was 60%, there was no regional wall motion abnormality, the chamber size was normal, and mild aortic regurgitation was observed. For the scheduled surgery, the pacemaker was reprogrammed to asynchronous AOO mode (80 bpm), and the pacemaker function and hemodynamics were tested by a certified technician and the attending cardiologist in the surgical holding area. 

In the operating room, balanced anesthesia with standard vital monitoring (i.e., ECG, arterial blood pressure (ABP), heart rate, peripheral oxygen saturation (SpO_2_), and capnography) was induced using propofol (Fresenius Kabi, Bad Homburg, Germany) (1.5 mg/kg) and rocuronium (Merck Sharp & Dohme Corp., Kenilworth, NJ, USA) (0.8 mg/kg), along with a remifentanil (Hanlim Pharm. Co., Ltd., Seoul, Korea) infusion. After tracheal intubation, anesthesia was maintained using desflurane (Baxter, Deerfield, IL, USA) with a continuous remifentanil infusion (0.05–0.1 mg/kg/min). For continuous arterial pressure monitoring, cannulation of the left radial artery was performed after a modified Allen’s test. There were no hemodynamic adverse events during the general anesthesia procedures (Table 1).

The IRE equipment used for the surgery was a NanoKnife system (AngioDynamics) and the surgeon planned to use five probes for localized prostate tissue ablation based on the MRI. During the procedure, the surgeon decided to use an additional probe (total of six probes for ablation) (Figure 2).

During the delivery of the first ablation pulse (as a pulse test for the correct placement of the needle), a sinus pause of 2.25 s that skipped two atrial paced beats was incidentally detected on the ECG, with no pulsation detected during continuous ABP or SpO_2_ monitoring. However, the atrial-paced rhythm recovered instantly and vital signs including ABP, heart rate, and SpO_2_ were acceptable. During the delivery of the second and third pulses, identical sinus pauses were observed due to failure to capture. Although the sinus pause recovered gradually, the more IRE pulses that were given, the longer the duration of the sinus pause (beyond 2.25 s). ABP during the pauses was lower than before the pause. We decided that the procedure should not continue and consulted the attending cardiologist to check pacemaker function. The technician and cardiologist inspected the pacemaker to determine whether there were any mechanical defects, and the pacemaker was reprogrammed from AOO (80 bpm) to DOO mode (80 bpm). The ablation was restarted, and even during pulse delivery there was no failure to beat, with a regular 80 bpm ventricular pacing rhythm observed on the ECG. The surgery was completed with no hemodynamic adverse events, including sinus pauses, and the patient was extubated and transported to the postanesthesia care unit (PACU). Vital signs were stable in the recovery phase. The cardiologist visited the PACU to examine the patient and reprogrammed the pacemaker to the previous mode (AAI). The patient was discharged on postoperative day three with no complications.

## 3. Discussion

The number of patients with a cardiac implantable electronic device (CIED) has continued to increase [13]. In the operating room, cardiac devices used in conjunction with emerging techniques, such as CIED, have presented a challenge for anesthesiologists, who must understand the basic principles and potential complications of such devices [14]. In particular, patients with pacemaker dependency may be vulnerable to EMI. IRE surgery is a risk factor for intraoperative EMI [15,16]. Although patients undergoing prostate IRE surgery may not require mandatory reprogramming of an asynchronous pacemaker due to the relatively large distance between the heart and prostate, the occurrence of intraoperative EMI may result in a hemodynamically fatal arrhythmia, such as a persistent electrical signal block, leading to impaired circulatory homeostasis in elderly patients [17].

In our case, 90 pulses, with a pulse length of 70 μs and within the target of 20–35 A, were delivered according to the IRE protocol. Atrial pacing failure occurred due to pulse delivery, resulting in transient asystole and contributing to decreased blood pressure. Although the actual mechanism of the pacing failure is not clear, it may have been due to myocardial tissue damage caused by pulse delivery and subsequent elevation of the pacing threshold in the right atrium [18]. This failure to beat may largely originate from the IRE pulse type and depend on the duration of delivery. As the right atrium is considered relatively more susceptible to sensing or pacing failure than the right ventricle, it is assumed to be more susceptible to EMI when operating in AOO mode during the procedure [19,20].

In addition, it should not be ignored that COVID-19 causes the occurrence of transient fatal arrhythmias through latent risk of cardiac vulnerability [21]. Although the patient had completed the IRE surgery safely and presented with no abnormal findings during the postoperative period, just by changing pacemaker mode the occurrence of arrhythmia in organs, which has little effect on the action of the IRE in organs far away from the heart, was observed in this case. This may be expected to be the trigger for the onset of arrhythmia from COVID-19 [22].

The intraoperative management of heart disease patients with a COVID-19 infection history (ongoing infectious phase or postinfectious recovery phase) has not been clearly determined yet, but COVID-19 infection may possibly contribute to the deterioration of the cardiac conduction system, such as sinus node disease or high degree AV block [23,24]. Cardiac electrophysiologic therapy has continuously played a reliable role to help COVID-19 patients recover from fatal arrhythmia despite the concerns of difficult lead positioning, health care providers safety, and the reduction of de novo implantation [25]. However, the relationship of the quality in pacemaker lead capture with a cardiac membrane that have been infected COVID-19 has not been fully investigated. However, we assume that irritable cardiac membrane due to inflammation or lagging response membrane due to fibrosis might possibly reduce the connection quality between a pacemaker lead pulse and cardiac membranous conduction.

IRE is considered a novel treatment option in oncology and has already been widely applied in clinical settings. Thus, it is necessary for anesthesiologists to understand the potential intraoperative complications of IRE surgery. In particular, in patients with a pacemaker scheduled for procedures that have a relatively high likelihood of EMI as well as a history of any infectious disease related to cardiac complication, meticulous cardiac monitoring is recommended in addition to the interrogation of the pacemaker. Close observation of ECG is required even when operating in asynchronous mode and it is necessary for clinicians to recognize the necessity of synchronization between IRE equipment and ECG where possible [26].

## 4. Conclusions

A multidisciplinary approach for perioperative management of patients should be emphasized for patient safety. It is difficult to determine the reasons and mechanisms of pacemaker failure unless one is familiar with the pacemaker, and the reasons for sinus pause are determined only retrospectively. Therefore, basically not only is communication among urologists, anesthesiologists, and cardiologists important, but preoperative evaluation should be more thorough to find causes quickly. This can increase the safety of patients with a pacemaker, who have a history of COVID-19 infections, undergoing procedures such as IRE for prostate cancer. 

## Figures and Tables

**Figure 1 medicina-58-01407-f001:**
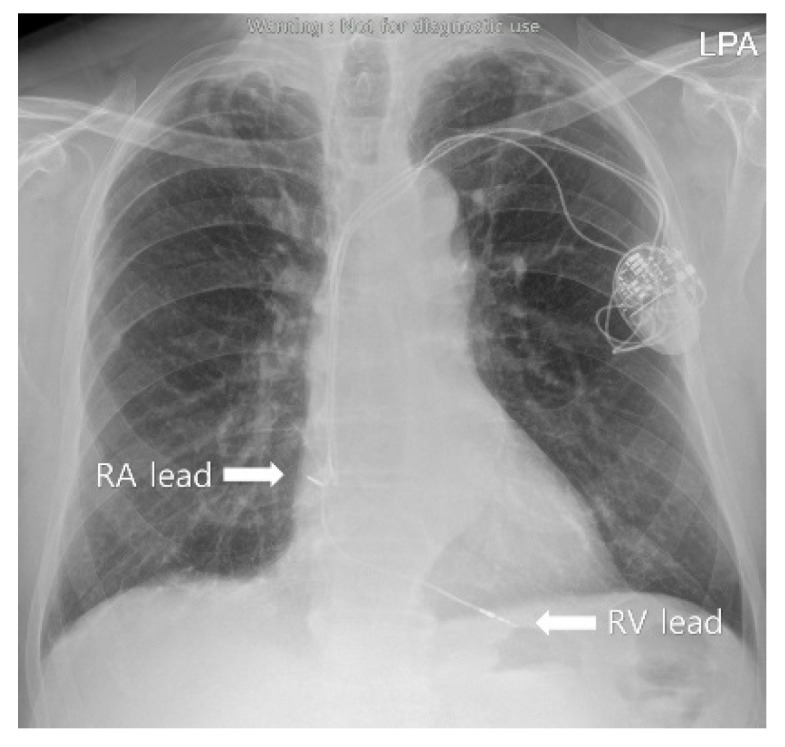
The preoperative evaluation chest X-ray.

**Figure 2 medicina-58-01407-f002:**
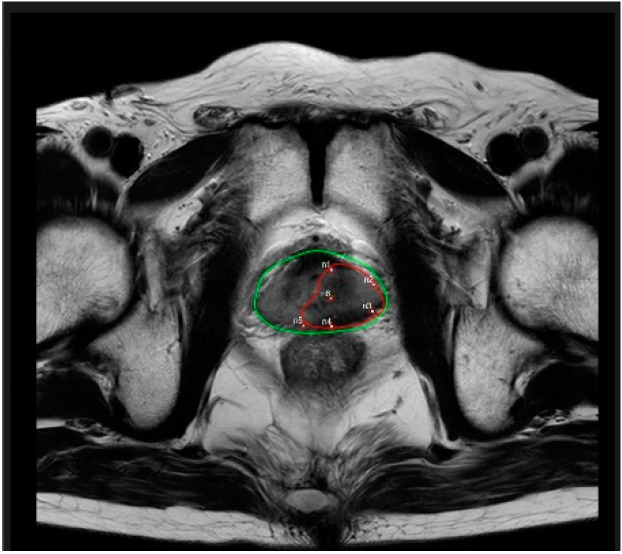
Magnetic resonance imaging showing the IRE ablation zone and probe locations. green line: probe location; red line: IRE zone.

**Table 1 medicina-58-01407-t001:** Hemodynamic parameters during intraoperative period.

Events	Induction	Beginning of Surgery	1st PD		2nd PD		3rd PD		Mode Change	4th PD	5th PD	End of Surgery
**Pacemaker Mode**	AOO									DOO		
**Hemodynamic Parameters**												
**ECG**	APR (Narrow QRS)	APR (Narrow QRS)	Sinus Pause	APR (Narrow QRS)	Sinus Pause	APR (Narrow QRS)	Sinus Pause	APR (Narrow QRS)	APR (Narrow QRS)	VPR (Wide QRS)	VPR (Wide QRS)	VPR (Wide QRS)
**SBP** (mmHg)	121	128	-	95	-	85	-	85	90	95	99	110
**DBP** (mmHg)	82	90	-	63	-	61	-	60	60	68	65	70
**HR** (beats/min)	80	80	-	80	-	80	-	80	80	80	80	80
**BIS**	54	50	32	34	32	35	32	34	40	46	45	40
**BT**	36.7	36.7	36.3	36.1	36.1	36.0	36.1	36.0	36.1	36.1	36.1	36.1
**SpO_2_** (%)	98	98	95	98	97	94	97	93	99	99	99	99
**ETCO_2_**	35	35	20	40	19	41	18	42	35	35	35	35

PD: pulse delivery; APR: atrial pacing rhythm; VPR: ventricular pacing rhythm. ECG: electrocardiography; SBP: systolic blood pressure; HR: heart rate; BIS: bispectral index; BT: body temperature; SpO_2:_: oxygen saturation; and ETCO_2_: end tidal CO_2_.

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
