# Peer review of "Sudden Occurrence of Pacemaker Capture Failure during Irreversible Electroporation Ablation for Prostate Cancer in Post-COVID-19 Patient: A Case Report"

_medicina, 2022, doi:10.3390/medicina58101407_

Round 1
Reviewer 1 Report
Abstract
- The abstract exceeds the number of words indicated by the journal.
- It is not methodologically correct for the keywords to be the same as the title. The purpose of keywords, plus the title of an article, is that they contribute to the visibility of your study.
- I recommend that authors explain the meaning of acronyms the first time they appear in the article. For example, electromagnetic interference (EMI)
Introduction
- The introduction clearly describes the phenomena related to Irreversible electroporation (IRE), magnetic resonance imaging (MRI) and electromagnetic interference (EMI) and their relationship, but the authors do not give a concise description about the general medical condition or relevant symptoms that will be discussed in the case report.
Case presentation
The case presentation, including all relevant demographic and descriptive information about the patient and a description of the symptoms, diagnosis, treatment, and outcome.
Discussion
The discussion provides adequate context and clearly explains the procedures performed for proper patient management. Despite the above, I believe that a more detailed explanation of the possible mechanism of arrhythmia induced by previous infection with COVID-19 in this patient is needed.
Conclusions
It is not methodologically correct to include citations in this section. I recommend that the authors remove citations 18 from the manuscript.
Author Response
Abstract
Point1: The abstract exceeds the number of words indicated by the journal.
Respond 1: we matched the number of words. (page 1 in the abstract section)
Point 2: It is not methodologically correct for the keywords to be the same as the title. The purpose of keywords, plus the title of an article, is that they contribute to the visibility of your study.
Respond 2: We modified the Keywords as you mentioned. (page 1 in the abstract section
Point3: I recommend that authors explain the meaning of acronyms the first time they appear in the article. For example, electromagnetic interference (EMI)
Respond 3: We described the meaning of acronyms the first time as you mentioned.
(page 1 in the abstract section, page 2 in the instruction section and page 3 in the case presentation section)
Introduction
Point4: The introduction clearly describes the phenomena related to Irreversible electroporation (IRE), magnetic resonance imaging (MRI) and electromagnetic interference (EMI) and their relationship,
but the authors do not give a concise description about the general medical condition or relevant symptoms that will be discussed in the case report.
Respond 4-1: We described patient’s condition or symptom concisely as you mentioned. (page 2 in the last sentence of introduction section)
[detailed explanation] the contents of the review overlapped and were presented in blue color phrases. (page 2 in the case presentation section)
Case presentation
The case presentation, including all relevant demographic and descriptive information about the patient and a description of the symptoms, diagnosis, treatment, and outcome.
Respond 4-2: We described patient’s preoperative history related to COVID-19 infection concisely. (page 3 in the case presentation section)
[detailed explanation] the contents of the review overlapped and were presented in blue color phrases.
Discussion
Point 5: The discussion provides adequate context and clearly explains the procedures performed for proper patient management. Despite the above,
I believe that a more detailed explanation of the possible mechanism of arrhythmia induced by previous infection with COVID-19 in this patient is needed.
Respond 5: We described more detailed explanation of that by considering your comments. (page 5 in the discussion section).
Conclusions
It is not methodologically correct to include citations in this section. I recommend that the authors remove citations 18 from the manuscript.
Respond 6: We deleted citation 18 as you mentioned.

Reviewer 2 Report
Sudden Occurrence of Pacemaker Capture Failure during Irreversible Electroporation Ablation for Prostate Cancer in Post COVID-19 Patient: A case report
Major comments:
“completely recovered” from COVID 19 – Possibility of blood tests for inflammatory markers? It is yet unknown how exactly COVID-19 damages the heart. Because of viruses that directly affect cardiomyocytes, systemic inflammatory reaction, endothelial injury and thrombotic inflammation, functional maladaptation of ACE-2 receptor-related pathways-COVID-19 may cause significant damage to the myocardium. My main issue is with the authors' assurance of patients' full recovery from COVID-19. Have you conducted any tests to confirm it? Even if patients who have recovered with COVID-19 can show positive RT-PCR results.
Does the IRE equipment generate enough EMI to be capable of interfering with pacemakers in general?
“identical sinus pauses” – Were subsequent pauses 2.25 seconds long, as the first one was? If this wasn’t the case, then how long did the sinus pause last as “more IRE pulses were given”?
It was long enough to initiate an “abrupt blood pressure decrease”. I’d assume this would be cause for greater concern.
“the occurrence of arrhythmia in organs away from the heart” – Isn’t an arrhythmia specific to the heart? Maybe irregular pulse or blood flow in other organs related to electrical signal interruptions locally?
Minor comments:
Please provide more demographic info about the patient
Please define other drugs listed there too, not only dutasteride as a 5alpha-reductase inhibitor.
In the conclusion – “a” should be capitalized in the first sentence.
Author Response
Point 1: “completely recovered” from COVID 19 – Possibility of blood tests for inflammatory markers?
Respond 1 : It means that the patient got vaccinated according to the vaccination schedule. For clarity, the phrase “completely recovered” was replaced to “had been recovered”.
[Detailed explanation] According to the practical guideline (in reference 12). we have determined ‘complete recovery of COVID-19 for elective surgery. Our multidisciplinary team determined his physical status was tolerable for elective IRE procedure based on the elective surgical timing protocol (beyond 7 weeks of a diagnosis of COVID-19 infection); the mild initial infection; no ongoing symptoms of COVID-19; comorbid and functional status; clinical priority and risk of prostate cancer progression; and complexity of IRE. The team did not recommend repetitive RT-PCR measures, because of possibly prolonged detection of COVID-19 particles (page 3 in the Case presentation section).
Point 2: It is yet unknown how exactly COVID-19 damages the heart. Because of viruses that directly affect cardiomyocytes, systemic inflammatory reaction, endothelial injury and thrombotic inflammation, functional maladaptation of ACE-2 receptor-related pathways-COVID-19 may cause significant damage to the myocardium.
Respond 2: As you mentioned, cardiac manifestation etiologies may be multifactorial that potentially involved in direct viral myocardial injury, hypoxia, hypotension, aggravating inflammation, ACE2-receptors downregulation, toxicity of COVID-19 treatment drug, and endogenous catecholamine adrenergic status.
Acute myocarditis and ventricular arrhythmia may be the first clinical signs of COVID-19, but in the Italian epidemic, many non-hospitalized patients without or with mild symptoms were found dead home while in quarantine. After hospital discharge without any respiratory or cardiac manifestations, the patients may have latent myocardial scars that result in atrial or ventricular fibrosis and, subsequently, increasing the propensity of cardiac arrhythmia (page 2 in the Introduction section).
Point 3: My main issue is with the authors' assurance of patients' full recovery from COVID-19. Have you conducted any tests to confirm it? Even if patients who have recovered with COVID-19 can show positive RT-PCR results.
Respond 3: We described the confirmation of recovery from COVID-19 in detail with reference in page 3.
[Detailed explanation] He had a COVID-19 related history. He had clinical COVID-19 suspicion with sore throat, mild fever (peak 37.3 ℃ with well acetaminophen control), cough, and finally showed the positive result of real time reverse transcription–polymerase chain reaction (RT–PCR). During home in quarantine, his infection course was tolerable without severe complications (such as dyspnea, hypoxia, and hypotension). After a week quarantine, he returned to daily life with mild muscle ache and fatigue and did not show major cardiac or respiratory sequelae. On a day before the IRE (post-COVID-19 infection day 60), our multidisciplinary team determined his physical status was tolerable for elective IRE procedure based on the elective surgical timing protocol (beyond 7 weeks of a diagnosis of COVID-19 infection); the mild initial infection; no ongoing symptoms of COVID-19; comorbid and functional status; clinical priority and risk of prostate cancer progression; and complexity of IRE. The team did not recommend repetitive RT-PCR measures, because of possibly prolonged detection of COVID-19 particles (Anaesthesia 2021 Jul;76(7):940-946. doi: 10.1111/anae.15464 in reference 12)
Point 4: Does the IRE equipment generate enough EMI to be capable of interfering with pacemakers in general?
Respond 4: Related description with reference [4,6,7] (page 2 in the Introduction section)
Point 5: “identical sinus pauses” – Were subsequent pauses 2.25 seconds long, as the first one was? If this wasn’t the case, then how long did the sinus pause last as “more IRE pulses were given”?
Respond 5: The subsequent pauses were not the same as the first one. That was described as “The more IRE pulses were given. The longer the duration of sinus pause”. We also described the duration of sinus pause after the first pause in manuscript.
[Detailed explanation] The device is applied to induce 90 pulses in sets of 10, with a short recharge period between each set. The duration of each pulse is 70 ms, separated by 100 ms, to reach a current between 20–40 A between the electrode pairs. Voltage is set according to the distance between the electrodes; however, the range is within 1200–1800 V/cm, with a maximum of 3000 V/cm. This cover range is suitable to warrant complete ablation without heat injury, while also avoiding undertreatment. The initial 10 pulses set as a “pulse test” to confirm for full muscle relaxation, as unexpected muscle spasm can move the position of the needles and shift the ablation area. The pulse test also establishes that the correct voltage is served between the needles and certifies that the treatment will not go beyond the thermal threshold. Adjustments are necessary course for optimizing the threshold, and then the remaining 80 pulses are given. (page 4 in the Case presentation).
Point 6: It was long enough to initiate an “abrupt blood pressure decrease”. I’d assume this would be cause for greater concern.
Respond 6: Yes, we agree with your comments. Because the degree of decrease was not increased and the mean arterial pressure did not fall when you calculated though the decrease in blood pressure was about 30% lower than blood pressure before the pause occurred. Therefore, we replaced “abrupt blood pressure decrease” to “ABP during the pauses was lower than before the pause” to convey the appropriate meaning. (page 4 in the Case presentation).
[Detailed explanation]
Because a pacemaker implanted patient with conduction disturbance was vulnerable for hemodynamic fluctuations, such as an abrupt hypotension, a multidisciplinary approach for perioperative management of patients should be emphasized for patient safety.
It is difficult to determine the reasons and mechanisms of pacemaker failure unless one is familiar with the pacemaker, and the reasons for sinus pause are determined only retrospectively. Therefore, basically, not only communication among urologists, anesthesiologists, and cardiologists is important, but thorough preoperative evaluation to quickly find the cause should have to be more careful. This can increase the safety of patient, who has a history of COVID-19 infection, with a pacemaker undergoing procedures such as IRE for prostate cancer (page 6 in the Case presentation).
Point 7: “the occurrence of arrhythmia in organs away from the heart” – Isn’t an arrhythmia specific to the heart? Maybe irregular pulse or blood flow in other organs related to electrical signal interruptions locally?
Respond 7: It is true that arrhythmia is specific to the heart. The irregular pulse or blood flow depends on the location of the EMI. In this article, we tried to emphasize that the location of the organ affected by EMI is a heart, which is an organ somewhat distant from the surgical site, and to suggest that one of the complications was an arrythmia.
[Detailed explanation]
We revied clearly to ‘the occurrence of arrhythmia, which has little effect on the action of the IRE in organs far away from the heart, was observed in this case. This may be expected to be the trigger for the onset of arrhythmia by COVID -19’ (page 5 in the Discussion section).
Minor comments:
Point 8: Please provide more demographic info about the patient
Respond 8: We described patient’s demographic information in detail according to your comment. (page 2 in the Case presentation).
Point 9: Please define other drugs listed there too, not only dutasteride as a 5 alpha-reductase inhibitor.
Respond 9: We described other drugs as dutasteride according to your comment. (page 3 in the Case presentation).
Point 10: In the conclusion – “a” should be capitalized in the first sentence.
Respond 10: We corrected “a” to “A” (page 6 in the Case presentation).

Round 2
Reviewer 2 Report
Thanks for the responses.